# Urinary prostaglandin metabolites as biomarkers for human labour: Insights into future predictors

**Eilidh M. Wood**[1], **Kylie K. Hornaday**[1], **Matthew Newton**[1], **Melinda Wang**[1],
**Stephen L. Wood**[2], **Donna M. Slater**[1,2]*

**1** Department of Physiology and Pharmacology, Cumming School of Medicine, University of Calgary, Calgary, Alberta, Canada, **2** Department of Obstetrics and Gynaecology, Cumming School of Medicine, University of Calgary, Calgary, Alberta, Canada

* dmslater@ucalgary.ca

## Abstract

Prostaglandins and other related molecules in the eicosanoid family have long been implicated in the process of both term and preterm labour. Although, exactly which eicosanoids are involved and whether they have utility as biomarkers for labour, remains to be shown. The objective of this study was to determine whether urinary prostaglandins and related molecules a) change with labour and/or cervical changes, at term and preterm, and/or b) are associated with timing of delivery in individuals with threatened preterm labour. Pregnant individuals were recruited into the following groups: $n = 32$ term non-labour, $n = 49$ term labour, $n = 15$ preterm non-labour controls, $n = 43$ threatened preterm labour with preterm delivery, and $n = 44$ threatened preterm labour with term delivery. Metabolites of prostaglandins $PGE_2$, $PGF_{2\alpha}$, $PGD_2$, and $PGI_2$ as well as 8-isoprostane were measured by ELISA. In addition, in a small ($n = 24$) subset of samples, 147 eicosanoids were measured using a mass-spectrometry based targeted lipidomics panel. At term labour prostaglandin $PGF_{2\alpha}$ and $PGE_2$ and $PGF_{2\alpha}$ metabolites were increased compared to term non-labour. There were no changes in any prostaglandin metabolites prior to labour onset. Prostaglandin $I_2$ metabolite was lower in individuals with threatened preterm labour who delivered preterm compared to those who went on to deliver at term. In our discovery cohort, we identified 20 additional eicosanoids as highly expressed in maternal urine, include members of the prostaglandin, hydroxyeicosatetraenoic acid (HETE), epoxyeicosatrienoic acid (EET), dihydroxy-octadecenoic acid (DiHOME), dihydroxy-eicosatrienoic acid (diHETrE), isoprostane, and nitro fatty acid eicosanoid families. In conclusion, we did not identify any prostaglandins that would have utility as predictors for term or preterm labour, however, we have identified diverse eicosanoids that have not been previously explored in the context of pregnancy and labour, highlighting novel areas for biomarker research.

**Data availability statement:** All relevant data are within the manuscript and its Supporting Information files.

**Funding:** This study was supported by the Canadian Institutes of Health Research (CIHR; https://cihr-irsc.gc.ca/e/193.html) to DMS and SLW (CIHR grant PJT-173295). The funders had no role in study design, data collection and analysis, decision to publish, or preparation of the manuscript.

**Competing interests:** The authors have declared that no competing interests exist.

## Introduction

Prostaglandins are biological mediators with key roles in reproduction, including pregnancy and parturition. Prostaglandins may contribute to the labour process by promotion of cervical ripening [1–3], uterine smooth muscle contraction [4,5] and membrane rupture [6–9]. In particular, the prostaglandins $PGE_2$ and $PGF_{2\alpha}$ and their synthetic analogues are clinically important in obstetrics and have been used to induce and augment labour since the late 1960s [10–12]. In addition, while not used clinically in obstetrics, $PGI_2$ and $PGD_2$ are also implicated in mechanisms that may play a role in human uterine relaxation and contraction [13–18]. Prostaglandins belong to a larger family of lipid signaling molecules called eicosanoids, and some of these related molecules, including the isoprostanes, which constitute a group of prostaglandin-like compounds, may also be involved in pregnancy and labour [19–22].

The mechanisms that contribute to the onset of human labour are not well understood. This limits our ability to clinically manage labour, most significantly in the case of preterm labour, which is a leading cause of neonatal morbidity and mortality worldwide [23]. Although many biomarkers for preterm labour have been proposed, none so far have sufficient predictive ability for clinical use [24,25]. Importantly, only 30–40% of individuals presenting with signs and symptoms of labour prior to 37 weeks gestation (i.e., threatened preterm labour) (TPTL), deliver prematurely [26,27], yet there is a lack of accurate measures for assessing which of these individuals will deliver preterm and which will deliver at term. A sensitive and specific biomarker for preterm delivery in the context of threatened preterm labour would allow for targeted clinical management and allocation of healthcare resources towards those at high risk of delivering preterm.

Available data on prostaglandin levels in maternal biofluids (urine, blood, amniotic fluid) provides some evidence that prostaglandin levels increase with labour [28], suggesting a possible role for prostaglandins as biomarkers for labour. Circulating prostaglandins are rapidly metabolized and secreted into the urine, which presents an opportunity to assess prostaglandin metabolites as potential biomarkers during pregnancy. Further, urine can be easily and non-invasively collected in relatively large volumes for biomarker analysis. We hypothesize that urinary prostaglandin levels are associated with labour and that possible alterations in prostaglandin levels associated with preterm labour are detectable in maternal urine. However, it is unknown whether urinary prostaglandin levels increase prior to labour onset and/or with the onset of true labour. A better understanding of how prostaglandins and/or their metabolites are involved in labour may allow for identification of biomarkers for preterm labour and preterm birth and possibly better targets for prevention and treatment of this condition. In addition, other, non-prostaglandin, eicosanoids may also be involved in the labour process, and a more complete understanding of the range of eicosanoids present in maternal circulation during pregnancy and labour may open new avenues for biomarker discovery.

Therefore, the first aim of this study is to determine whether the urinary prostaglandins $PGF_{2\alpha}$ and prostaglandin metabolites PGFM, PGEM, PGIM, and tetranor-PGDM

(t-PDGM), as well as 8-isoprostane, a) change with labour and/or cervical changes, at term and preterm, and b) are associated with timing of delivery in those with TPTL. The second aim of this study is to perform a discovery analysis on a subset of samples using a mass-spectrometry-based eicosanoid panel and determine if any additional eicosanoids are associated with labour.

## Methods

### Participants and sample collection

Institutional ethics was obtained for the study (University of Calgary Conjoint Health Research Ethics Board, REB18–0648, date of issue: May 7, 2018) and all participants provided written consent for collection of urine and medical record data on mode of delivery and pregnancy outcomes. Participants were recruited at the Foothills Medical Centre (Calgary, AB) between May 2018 and July 2020. At recruitment, participants were classified based on gestational age [term (37–42 weeks) or preterm (<37 weeks)] and presence or absence of labour. Once delivered, participants were categorized into the following groups: term non-labour (TNL), n = 32; term labour (TL), n = 49, preterm non-labour controls (PTNL), n = 15; threatened preterm labour with preterm delivery (TPTL-PTD), n = 43, or threatened preterm labour with term delivery (TPTL-TD), n = 44 (Fig 1).

Term non-labour participants were recruited at presentation for induction or caesarean section at ≥37 weeks gestation with no symptoms of labour (no uterine contractions). Term labour was defined as spontaneous onset of labour, characterized by uterine contractions and cervical dilation, leading to delivery between 37–42 weeks gestation. Preterm non-labour controls were recruited either at presentation for non-delivery related reasons (n = 12/15) or at presentation for induction or caesarean section at <37 weeks with no symptoms of labour (intact membranes and no reported contractions) (n = 3/15). Threatened preterm labour was diagnosed based on clinical assessment by the attending physician and defined as presentation with symptoms of labour including spontaneous onset of uterine contractions or cramping, cervical changes including shortening and/or dilation, and/or premature rupture of membranes, prior to 37 weeks gestation. Participants with threatened preterm labour followed by a medically indicated preterm delivery, defined as delivery following induction of labour or caesarean section for fetal or maternal indications not related to labour, were excluded from the analysis. Deliveries were medically indicated for the following reasons: fetal heart rate abnormality, urinary retention, uterine

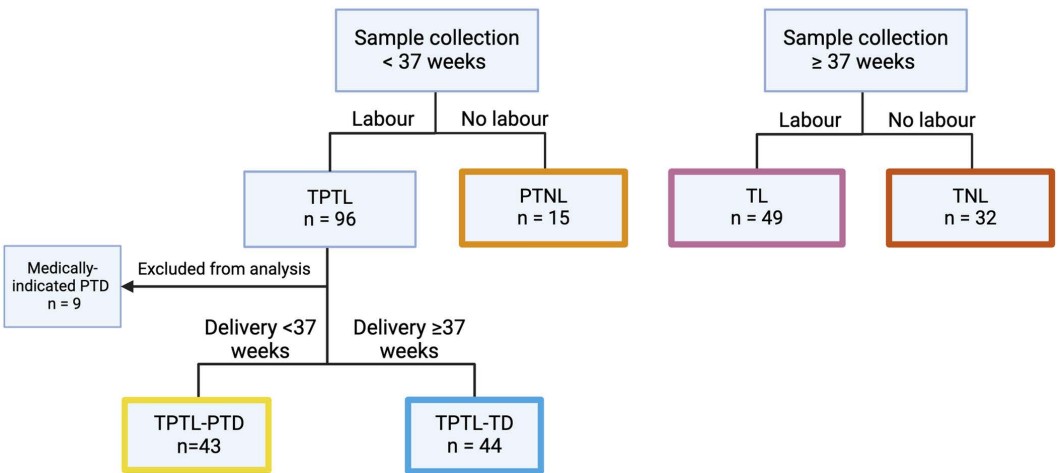

**Fig 1. Participant grouping.** Based on gestational age at sample collection (term vs preterm), presence or absence of labour, and outcome (preterm vs term delivery). TPTL = threatened preterm labour, PTNL = preterm non-labour controls, PTD = preterm delivery, TD = term delivery, TL = term labour, TNL = term non-labour. Created in BioRender. Wood, E. (2024) https://BioRender.com/i35i656.

dehiscence, and maternal medical conditions unrelated to pregnancy. An obstetrician blinded to the assay results was consulted to review cases where the outcome was unclear. Urine samples were collected in sterile containers at the time of recruitment and frozen at −80°C prior to analysis. Clinical information including cervical dilation and length, duration of ruptured membranes, and duration of contractions were recorded on study datasheets at the time of sample collection. Cervical score was calculated using the equation:

$$Cervical\ score\ =\ cervical\ length\ (cm) - cervical\ dilation\ (cm)$$

and used as a measure of cervical ripening and proximity to spontaneous delivery, as it has been demonstrated to perform similarly to the Bishop Score in predicting spontaneous delivery [29]. A cervical score of −10 indicates a fully dilated, fully effaced cervix, while a cervical score of 4 indicates a long and closed (i.e., unfavourable) cervix.

### Enzyme-linked immunosorbent assay (ELISA)

Urinary concentrations of 13,14-dihydro-15-keto $PGF_{2\alpha}$ (prostaglandin F metabolite; PGFM), tetranor-PGDM (t-PGDM), prostaglandin I metabolites (PGIM; 2,3-dinor-6-keto Prostaglandin $F_{1\alpha}$ and 20-carboxy-2,3-dinor-6-keto Prostaglandin $F_{1\alpha}$), and 8-isoprostane were measured by ELISA kit (Cayman Chemical, Ann Arbor, MI, item numbers #516671, #501001, #501100, and #516351, respectively). Prostaglandin E2 metabolite (PGEM) was measured using a PGE Metabolite ELISA kit (Cayman Chemical, Ann Arbor, MI, item number #514531) that converts the major circulating and urinary metabolites of PGE2, 13,14-dihydro-15-keto $PGA_2$ and 13,14-dihydro-15-keto $PGE_2$, to a single stable derivative that can be quantified by ELISA. Samples were run in duplicate, at dilutions of between 1:2–1:50 for PGFM, 1:5–1:50 for t-PGDM, 1:50 for PGIM, 1:25–1:50 for 8-isoprostane and 1:50–1:100 for PGEM. All samples were measured in duplicate. Assays were performed according to manufacturer's protocol. Samples were assayed in random order and blinded to pregnancy outcome. ELISA data for PGFM, PGEM, t-PDGM, 8-isoprostane, PGIM and $PGF_{2\alpha}$ is provided in S1 Appendix.

### Urinary concentration correction

Urine concentration was normalized to specific gravity using the Boeniger method [30,31]:

$$PG_{sg} = PG * \frac{1.012 - 1}{SG - 1}$$

Where $PG_{sg}$ is the specific gravity corrected prostaglandin metabolite concentration, 1.012 is the median specific gravity among all samples analyzed, PG is the measured prostaglandin concentration in pg/mL, and SG is the specific gravity of the sample. Specific gravity was measured using a Laxco™ Handheld Analog Clinical Refractometer (Thermo Fisher Scientific, Ottawa, ON, catalogue no. IRC200ATC) according to the manufacturer's instructions.

Creatinine levels were measured for urine concentration normalization, however, due to a weak correlation between creatinine and gestational age (Spearman's rank correlation: r(181)=0.31, p<0.0001; S2 Appendix), all subsequent analysis was conducted on specific gravity-corrected values.

### Reverse phase ultra high-performance liquid chromatography-mass spectrometry

A subset of n=24 urine samples were selected for a discovery analysis using a mass-spectrometry-based eicosanoid panel. Lipid analysis was performed at the University of California San Diego (UCSD) Lipidomics Core using a mass spectrometry-based comprehensive eicosanoid panel. The eicosanoid panel consists of 147 lipids including, but not limited to, all major prostaglandins and their metabolites, four fatty acids (arachidonic acid, AA; eicosapentaenoic acid, EPA; docosahexaenoic acid, DHA; adrenic acid), isoprostanes, leukotrienes, lipoxins, resolvins, maresins, protectins,

hydroxyeicosatetraenoic acids (HETEs), hydroxyeicosapentaenoic acids (HEPEs), hydroxydocosahexaenoic acids (HDoHEs), hydroxyoctadecatrienoic acids (HOTrEs), hydroxyeicosatrienoic acids (HETrEs), hydroxyoctadecadienoic acids (HODEs), epoxyeicosatrienoic acids (EETs), epoxydocosapentaenoic acids (EpDPEs), epoxyoctadecenoic acids (EpOMEs), and nitrated fatty acids. Full details of the eicosanoids measured in the panel are provided in S3 Appendix. Analysis was performed as described by Quehenberger and colleagues [32]. Briefly, eicosanoids were isolated from 400 µL urine supplemented with 100 µL internal standard mix by solid phase extraction using strata-x polymeric reverse phase columns (8B-S100-UBJ Phenomenex). Samples were then evaporated and reconstituted in 50 µL of buffer (63% H2O, 37% acetonitrile, 0.02% acetic acid) and separated by reverse phase ultra-high performance liquid chromatography on an ACQUITY UPLC System Waters BEH-Shield column (2.1 x 100 mm, 1.7µM; Waters, Milford, MA, USA). Eicosanoids were analyzed using a triple quadrupole linear ion trap mass spectrometer (Sciex 6500 Qtrap). Eicosanoids were quantitated by comparison with standards. Data analysis was performed using Analyst and MultiQuant software (Applied Biosystems). Mass spectrometry data is provided in S1 Appendix.

### Statistics and data presentation

Distributions of variables were assessed for normality using the Shapiro-Wilk test. Prostaglandin levels were corrected for urinary concentration by specific gravity and log2-transformed for normality. Differences in prostaglandin levels between groups were assessed by t-test. To assess the association between prostaglandin levels and cervical ripening the TL and TNL groups were combined and the PTNL, TPTL-PTD, and TPTL-TD groups were combined. Associations between prostaglandin levels and cervical score were assessed by Spearman's correlation. Associations between prostaglandin levels and days from collection to delivery were assessed by linear regression. Data are presented as mean ± standard deviation (sd) for data normally distributed data and median ± interquartile range (IQR) for non-parametric data. Differences were considered significant with a 2-tailed p-value<0.05. Statistical analysis of demographic information was performed using R version 4.1.3. All other statistical analysis and visualization was performed using GraphPad Prism 9 software. Due to the small sample size of the discovery cohort combined with a number of eicosanoid levels below the limit of detection, statistical analysis was not performed on discovery cohort eicosanoid data.

## Results

### Participant demographics

Demographic and clinical characteristics of participants are shown in Table 1. Compared to the TNL group, participants in the TL group had later median gestational age at collection (39.7 vs 38.6 weeks; p<0.001) and later median gestational age at delivery (39.7 vs 38.6 weeks; p<0.001). The mean maternal age of participants in the PTNL (mean=31±5.24 years; p=0.030), TPTL-PTD (mean=31.7±4.88 years; p=0.022), and TPTL-TD (mean=31.3±5.39 years; p=0.009) groups, was lower than the TNL group (mean age=34.5±5.47 years).

### Urinary prostaglandin levels with labour and gestation

The TL group was associated with significantly higher levels of PGFM, PGEM and PGF$_{2\alpha}$ compared to the TNL group (Table 2 and Fig 2). The levels of PGIM were significantly higher in TPTL-TD compared to the TPTL-PTD group. No other prostaglandin levels were different between TPTL-TD and TPTL-PTD groups and no gestational age differences in prostaglandin levels were observed (Fig 2). Within the TPTL-PTD group, n=4 participants had PPROM, therefore an additional, separate, analysis was conducted with these participants excluded. The only difference observed in this analysis was for PGIM, which become non-significant with the removal of the PPROMs (p=0.071 vs p=0.036) (see S4 Appendix).

**Table 1. Demographic and clinical characteristics of participants.**

| | PTNL (n = 15) | TNL (n = 32) | TL (n = 49) | TPTL-PTD (n = 43) | TPTL-TD (n = 44) | p-value |
|---|---|---|---|---|---|---|
| **Gestational age at collection (days)** | | | | | | |
| Median (IQR) | 223.0 (194.5-247.0) | 270.0 (267.5-273.0) | 278.0 (273.0-282.0) | 211.0 (182.0-247.0) | 220.0 (196.0-235.2) | TNL vs TL: <0.001 PTNL vs TPTL-PTD vs TPTL-TD: >0.05 |
| **Gestational age at delivery (days)** | | | | | | |
| Median (IQR) | 262 (256.5-273) | 270.0 (268.8-273.0) | 278.0 (274.0-282.0) | 219.0 (187.5-254.0) | 272.0 (266.7-275.0) | TNL vs TL: <0.001 TNL vs PTNL: >0.05 TNL vs TPTL-TD: >0.05 TL vs PTNL: <0.001 TL vs TPTL-TD: <0.001 PTNL vs TPTL-TD: 0.027 |
| **Time from collection to delivery (days)** | | | | | | |
| Median (IQR) | 33.5 (1-64.2) | 0 (0-1) | 0 (0−0) | 1 (0-5.5) | 55 (30.8-78.5) | NA |
| **Maternal age** | | | | | | |
| Mean ± sd | 31 ± 5.24 | 34.5 ± 5.47 | 32.3 ± 4.98 | 31.7 ± 4.88 | 31.3 ± 5.39 | TNL vs PTNL: 0.030 TNL vs TPTL-PTD: 0.022 TNL vs TPTL-TD: 0.009 |

Differences in maternal age between groups was assessed by ANOVA followed by pairwise t-test. Differences in all other variables were assessed by Wilcoxon rank sum test (no corrections for multiple testing). TNL = term non-labour, TL = term labour, PTNL = preterm non-labour controls, TPTL-PTD = threatened preterm labour – preterm delivery, TPTL-TD = threatened preterm labour – term delivery.

### Urinary prostaglandin levels and cervical ripening

At term, PGEM levels were positively associated with cervical ripening ($R^2 = 0.069$, $\beta = -0.099$, $p = 0.033$; Fig 3a), while levels of PGFM, t-PGDM, 8-isoprostane, PGIM, and $PGF_{2\alpha}$ showed no changes (S5 Appendix, Supplemental Fig 2a-e). At preterm, t-PGDM and PGIM levels were negatively associated with cervical ripening (t-PGDM: $R^2 = 0.081$, $\beta = 0.11$ $p = 0.017$; PGIM: $R^2 = 0.070$, $\beta = 0.12$, $p = 0.035$; Fig 3b, c), while PGFM, PGEM, 8-isoprostane, and $PGF_{2\alpha}$ showed no association (S5 Appendix, Supplemental Fig 2f-I).

### Prostaglandin levels and time to delivery

In linear regression models of the associations between urinary prostaglandin levels and the number of days between sample collection and delivery in participants with threatened PTL, PGFM ($R^2 = 0.046$, $\beta = -0.008$, $p = 0.048$), 8-isoprostane ($R^2 = 0.074$, $\beta = -0.015$, $p = 0.017$), and $PGF_{2\alpha}$ ($R^2 = 0.065$, $\beta = -0.010$, $p = 0.022$) levels were significantly associated with decreasing time to delivery, and PGIM levels were associated with increasing time to delivery ($R^2 = 0.062$, $\beta = 0.010$, $p = 0.024$; Fig 4).

### Discovery cohort eicosanoid profile

To determine additional eicosanoids that may be present in maternal urine, a subset of n = 24 urine samples were selected for a discovery analysis using a mass-spectrometry-based eicosanoid panel. Groups were composed of n = 6 threatened preterm labour with preterm delivery (TPTL-PTD), n = 6 preterm non-labour controls (PTNL), n = 6 term labour (TL), and n = 6 term non-labour (TNL) (Table 3). Of the 147 eicosanoids measured, 66 were not detected in any sample. A total of 81/147 eicosanoids and fatty acids were detected in at least one urine sample and ranged from concentrations of 0.01 pmol/mL to 1016.17 pmol/mL. Of the 81 eicosanoids, 4 were detected in all samples (9,10-diHOME, 9,10-EpOME, 9-HODE, and 13-HODE) and 20 were detected in at least 50% of samples (Table 4). Eicosanoids detected in at least 50%

**Table 2. Specific gravity corrected and log base 2 transformed urinary prostaglandin levels by group.**

| | TNL (n=32) | TL (n=49) | PTNL (n=15) | TPTL-PTD (n=43) | TPTL-TD (n=44) | p-value |
|---|---|---|---|---|---|---|
| **PGFM** | | | | | | |
| Mean±sd | 11.4±0.799 | 12.2±1.17 | 10.9±0.805 | 10.5±1.29 | 10.3±1.23 | TL vs TNL: 0.003 |
| Total missing (%) | 0 (0%) | 0 (0%) | 0 (0%) | 1 (2.3%) | 0 (0%) | TPTL-PTD vs TPTL-TD: >0.05 PTNL vs TNL: 0.023 |
| **PGEM** | | | | | | |
| Mean±sd Total | 9.79±1.03 | 10.5±1.06 | 9.51±0.933 | 10.2±1.57 | 10.0±1.21 | TL vs TNL: 0.001 TPTL-PTD vs TPTL-TD: >0.05 |
| missing (%) | 0 (0%) | 1 (2.0%) | 0 (0%) | 0 (0%) | 1 (2.3%) | PTNL vs TNL: >0.05 |
| **t-PGDM** | | | | | | |
| Mean±sd | 13.2±0.790 | 13.1±0.846 | 13.0±0.755 | 12.5±0.955 | 12.7±1.11 | All p>0.05 |
| Total missing (%) | 0 (0%) | 0 (0%) | 0 (0%) | 0 (0%) | 0 (0%) | |
| **8-isoprostane** | | | | | | |
| Mean±sd | 11.1±1.52 | 10.8±1.42 | 11.0±2.29 | 10.8±1.88 | 10.1±1.52 | All p>0.05 |
| Total missing (%) | 4 (12.5%) | 2 (4.1%) | 0 (0%) | 5 (11.6%) | 5 (11.4%) | |
| **PGIM** | | | | | | |
| Mean±sd | 14.5±0.849 | 14.4±0.965 | 14.6±1.39 | 14.4±1.12 | 15.0±1.25 | TL vs TNL: >0.05 |
| Total missing (%) | 3 (9.4%) | 2 (4.1%) | 0 (0%) | 2 (4.7%) | 2 (4.5%) | TPTL-PTD vs TPTL-TD: 0.036 PTNL vs TNL: >0.05 |
| **PGF$_{2\alpha}$** | | | | | | |
| Mean±sd | 10.3±0.703 | 10.8±0.786 | 10.5±0.709 | 10.6±1.37 | 10.6±1.00 | TL vs TNL: 0.009 |
| Total missing (%) | 5 (15.6%) | 2 (4.1%) | 1 (6.7%) | 2 (4.7%) | 5 (11.4%) | TPTL-PTD vs TPTL-TD: >0.05 PTNL vs TNL: >0.05 |

TNL=term non-labour, TL=term labour, PTNL=preterm non-labour controls, TPTL-PTD=threatened preterm labour – preterm delivery, TPTL-TD=threatened preterm labour – term delivery. Differences analyzed by t-test.

of samples include members of the prostaglandin, hydroxyeicosatetraenoic acid (HETE), epoxyeicosatrienoic acid (EET), dihydroxy-octadecenoic acid (DiHOME), dihydroxy-eicosatrienoic acid (diHETrE), isoprostane, and nitro fatty acid eicosanoid families.

## Discussion

Here we have shown that metabolites of prostaglandins F$_{2\alpha}$ and E$_2$, PGFM and PGEM, are increased in maternal urine with term labour compared to term non-labour. However, as evidenced by the lack of gestational age-related changes in prostaglandin metabolite levels, none of the prostaglandins measured here appear to increase leading up to labour onset. Additionally, we show that PGIM was the only metabolite that displayed a significant, albeit small, difference in individuals with threatened PTL who delivered preterm compared to those who delivered at term. The evidence provided here therefore suggests that none of the prostaglandin biomarkers measured are likely to be *clinically* useful for prediction of labour onset, either at term or in the instance of threatened preterm labour.

### PGFM and PGEM levels are higher with term labour, but do not change prior to labour

That prostaglandins PGF$_{2\alpha}$ and PGE$_2$ and their metabolites increase during labour is well established in the literature and has been consistently demonstrated in various biofluids including amniotic fluid [19,33–44], maternal blood [38,45–53], and maternal urine [38,54–56]. That we were not able to find any changes in prostaglandin levels with gestational age (preterm non-labour vs term non-labour), is also generally consistent with previous studies that have measured

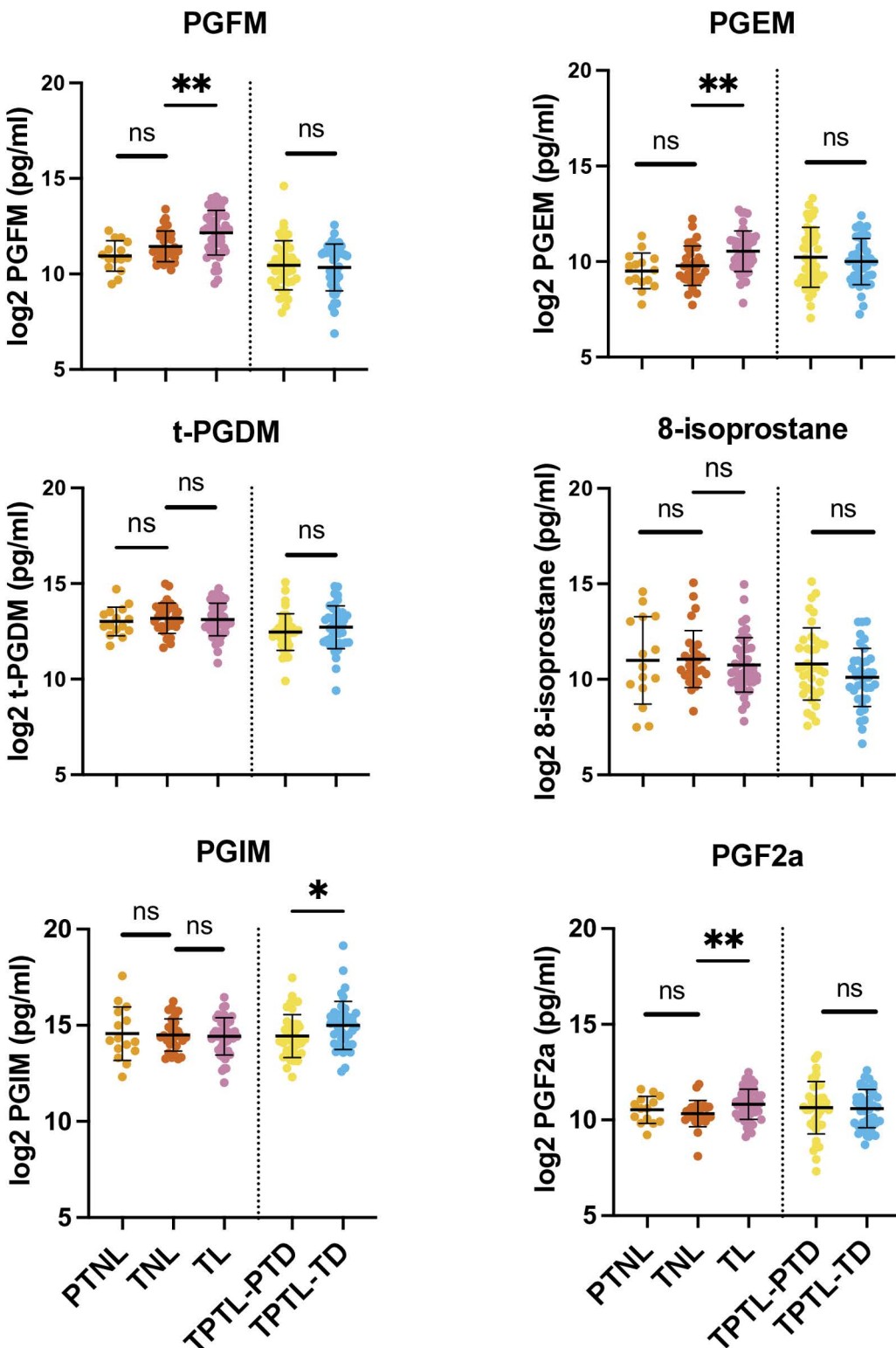

**Fig 2. Prostaglandin and metabolite levels in maternal urine with and without labour.** Prostaglandin levels were log2 transformed prior to statistical analysis. Differences analyzed by t-test. TNL = term non-labour, TL = term labour, PTNL = preterm non-labour controls, TPTL-TD = threatened preterm labour–term delivery, TPTL-PTD = threatened preterm labour-preterm delivery. *p < 0.05, **p < 0.01, ns = non-significant.

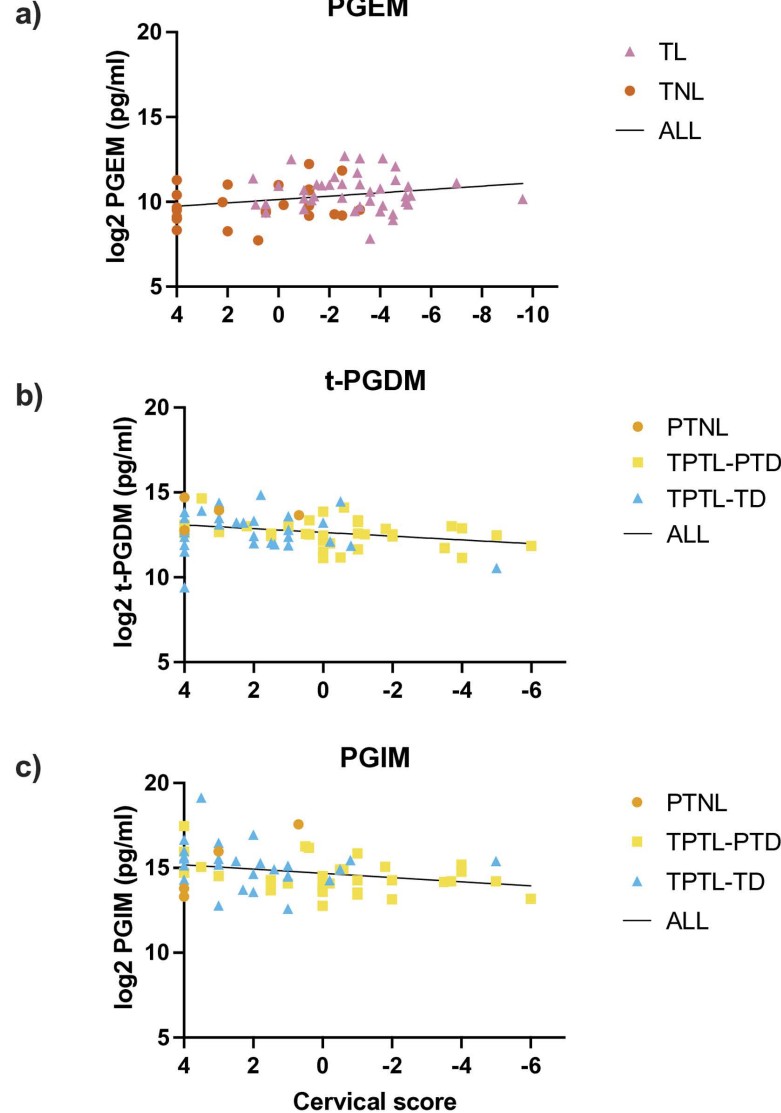

**Fig 3. Association between urinary prostaglandin levels and cervical score.** Analyzed by linear regression. a) PGEM: $R^2 = 0.069$, $\beta = -0.099$, $p = 0.034$; b) t-PGDM: $R^2 = 0.081$, $\beta = 0.11$, $p = 0.017$; c) PGIM: $R^2 = 0.070$, $\beta = 0.12$, $p = 0.035$. TNL = term non-labour, TL = term labour, PTNL = preterm non-labour controls, TPTL-TD = threatened preterm labour–term delivery, TPTL-PTD = threatened preterm labour-preterm delivery. Regression equations represent all samples grouped together.

prostaglandin metabolites in maternal blood [38,48,57–60]. Several studies conducted using amniotic fluid samples, however, have reported increases in $PGF_{2\alpha}$ and $PGE_2$ around term, prior to labour onset [33,34,41,42,61,62], suggesting that biomarkers predictive of labour onset may be present in this biofluid. Although maternal urine is appealing as a minimally invasive biomarker option, it may not provide the best reflection of the physiological processes occurring prior to labour. It may be possible that uterine prostaglandin production does increase prior to labour onset, and that urinary output of prostaglandin metabolites simply does not reflect this increase. Although this seems unlikely to be the case, as previous studies have demonstrated that increases in prostaglandin metabolites show up rapidly (i.e., at least within 2 hours) in urine following systemic increases in these prostaglandins [63,64]. An alternative explanation is that $PGF_{2\alpha}$ and $PGE_2$

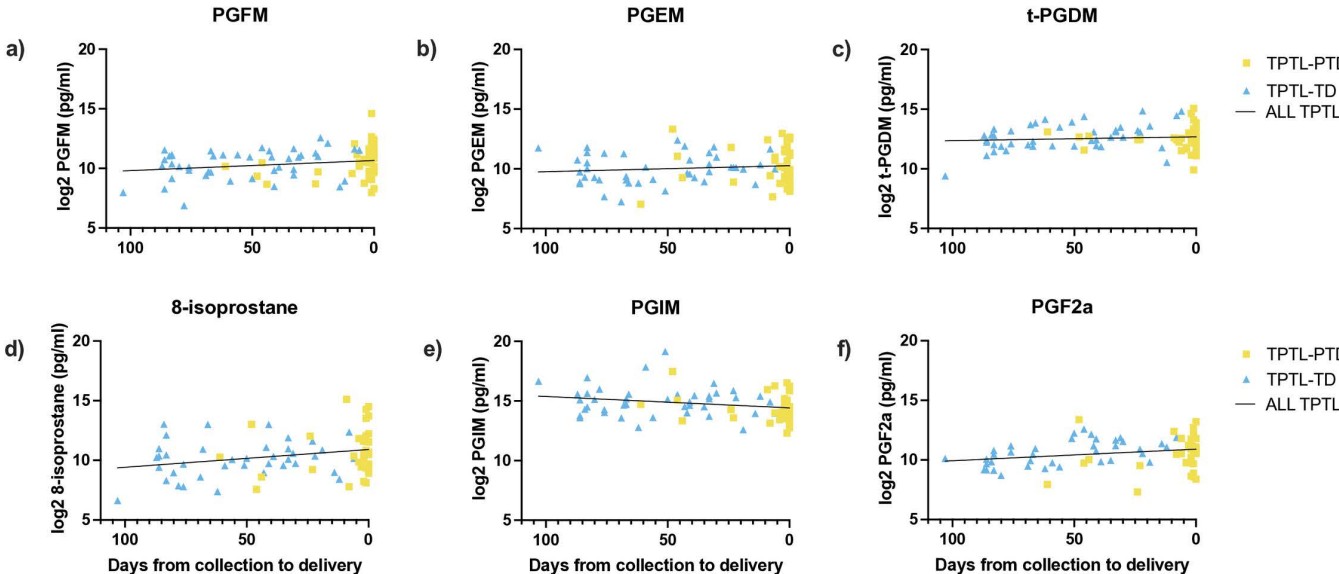

**Fig 4. Association between urinary prostaglandin levels and timing to delivery in individuals with threatened preterm labour.** Analyzed by linear regression. a) PGFM: $R^2 = 0.046$, $\beta = -0.008$, $p = 0.048$; b) PGEM: $R^2 = 0.013$, $\beta = -0.005$, $p = 0.30$; c) t-PGDM: $R^2 = 0.009$, $\beta = -0.003$, $p = 0.37$; d) 8-isoprostane: $R^2 = 0.074$, $\beta = -0.015$, $p = 0.017$; e) PGIM: $R^2 = 0.062$, $\beta = 0.010$, $p = 0.024$; f) PGF2a: $R^2 = 0.065$, $\beta = -0.010$, $p = 0.022$. PTNL = preterm non-labour controls, TPTL-TD = threatened preterm labour–term delivery, TPTL-PTD = threatened preterm labour-preterm delivery.

**Table 3. Discovery cohort demographics.**

|  | PTNL (n = 6) | TPTL-PTD (n = 6) | TNL (n = 6) | TL (n = 6) | p-value |
|---|---|---|---|---|---|
| **Gestational age at collection (weeks)** | 30.1 (26-34.4) | 29 (26.4-33.5) | 39.3 (38.6-40.4) | 39.1 (39-39.6) | TNL vs TL > 0.05 PTNL vs TPTL-PTD > 0.05 |
| **Gestational age at delivery (weeks)** | 39.1 (38.6-40.5) | 29.2 (26.6-33.5) | 39.3 (38.6-40.4) | 39.1 (39-39.6) | TNL vs TL > 0.05 TNL vs PTNL > 0.05 TL vs PTNL > 0.05 |
| **Time from collection to delivery (days)** | 58 (38.8-93) | 1.5 (0.25-2) | 0 (0-0.75) | 0 (0−0) | TNL vs TL > 0.05 |
| **Maternal age** | 32 (27.5-35) | 30 (22-35.8) | 36 (30.5-39.2) | 31.5 (28.8-33.5) | All > 0.05 |

Data represented as median (IQR). TNL = term non-labour, TL = term labour, PTNL = preterm non-labour control, TPTL-PTD = threatened preterm labour – preterm delivery, NR = not reported, NA = not applicable.

metabolites do increase with term labour, but do not change prior to labour onset, which could support the premise that prostaglandin metabolite production is a consequence of labour onset, rather than a trigger for this process, as suggested by others [65].

## Prostaglandins and cervical ripening

We also noted PGEM levels were associated with cervical ripening at term, but not preterm. There is extensive literature detailing the association between exogenous administration of $PGE_2$ and cervical ripening [66,67], however few others have investigated the association between endogenous levels of $PGE_2$ or its metabolites, and degree of cervical ripening [68]. The results demonstrated here provide support for a role for $PGE_2$ in the physiological process of cervical ripening, although the question of whether the increase in $PGE_2$ metabolites is a cause or a consequence of cervical ripening remains.

**Table 4. Eicosanoids with ≥50% of samples above limit of detection in discovery cohort.**

| Eicosanoid | >LOD (%) |
|---|---|
| $PGF_{2\alpha}$ | 92 |
| $PGE_2$ | 75 |
| tetranor-PGFM | 58 |
| tetranor-PGEM | 54 |
| tetranor 12-HETE | 96 |
| $PGA_2$ | 67 |
| $PGB_2$ | 54 |
| $PGJ_2$ | 54 |
| 5-HETE | 50 |
| 13-HODE | 100 |
| 9-HODE | 100 |
| 11,12-EET | 50 |
| 9,10-EpOME | 100 |
| 12,13-EpOMe | 83 |
| 8,9-diHETrE | 54 |
| 9,10-diHOME | 100 |
| 12,13-diHOME | 96 |
| Arachidonic acid | 88 |
| 2,3 dinor 8-iso $PGF_{2\alpha}$ | 96 |
| 9-Nitrooleate | 88 |

LOD = limit of detection.

### Prostaglandin metabolite levels may not be useful as predictors of preterm delivery in individuals with threatened PTL

Of special interest in this study was whether prostaglandin metabolite levels could be used to differentiate between threatened preterm labour that resolves and eventually ends in a term delivery from threatened preterm labour that results in a preterm delivery. Previous studies have identified an association between higher levels of $PGF_{2\alpha}$ in amniotic fluid and shorter time to delivery in individuals with preterm labour [69] and PPROM [70]. In addition, Peiris et al. found higher levels of amniotic fluid $PGE_2$ and PGFM in individuals with preterm labour and intra-amniotic infection or intra-amniotic inflammation who delivered preterm compared to individuals with preterm labour who delivered at term, although there were no differences in prostaglandins or metabolites when comparing preterm labour without inflammation [71]. We did not find any differences in urinary $PGE_2$ or $PGF_{2\alpha}$ metabolite levels between those with threatened PTL and preterm delivery compared to those with threatened PTL and term delivery, however, we did find higher levels of prostaglandin $I_2$ metabolites (PGIM) in TPTL-TD compared to TPTL-PTD. Prostaglandin $I_2$ ($PGI_2$) is known to induce smooth muscle relaxation in the uterus [15,16], and has been implicated in pregnancy maintenance [72]. It is therefore possible that lower levels of $PGI_2$ could contribute to labour onset via loss of quiescent signaling in the uterus, although we did not see any changes in PGIM prior to labour onset when comparing PTNL to TNL. In our analysis of the association between PGIM and timing to delivery in individuals with threatened PTL, PGIM accounted for a small percentage ($R^2 = 0.062$) of the variance in time to delivery, making this metabolite unlikely to be useful as a biomarker for preterm delivery in individuals with threatened PTL.

### Role for diverse eicosanoids in labour

Beyond prostaglandins there is limited information on the levels and trajectories of eicosanoids throughout pregnancy and a poor understanding of how these molecules may contribute to the process of labour onset. In the present investigation,

we have identified several non-prostaglandin eicosanoids present in maternal urine at levels comparable to, or higher than, maternal urinary prostaglandin levels, some of which could potentially be useful as biomarkers for labour. Among the eicosanoids identified, include members of the lipoxygenase (LOX)-derived HETEs and CYP450-derived EETs and diHOMEs. Notably, previous investigations have reported that higher plasma levels of the LOX-derived eicosanoid, 5-HETE, in the second trimester are associated with increased risk of spontaneous preterm birth [73,74]. Aung and colleagues also reported associations between plasma levels of 12,13-dihydroxy-octadecenoic acid (12,13-diHOME) and 9,10-dihydroxy-octadecenoic acid (9,10-diHOME) and increased risk of preterm birth and found that LOX and CYP450 pathway eicosanoids demonstrated better predictive capability for spontaneous preterm birth than COX pathway eicosanoids (i.e., prostaglandins) [73]. Further, Borkowski et al. report an association between spontaneous preterm birth and LOX pathway metabolites in obese pregnant individuals, but not in "normal weight" pregnancies, suggesting that this association may be BMI-dependent [75]. Further investigation into the role of these eicosanoids in pregnancy and labour, and other factors that may influence their levels, could potentially uncover better biomarkers for conditions like preterm birth.

## Limitations

In this analysis, we included participants with preterm labour and participants with preterm premature rupture of membranes (PPROM) in one group under threatened PTL. Although both preterm labour and PPROM are events that contribute to spontaneous delivery, it is possible that the molecular pathways leading to the onset of each of these events is distinct. In the future, studies with larger numbers should aim to analyse these groups separately.

## Conclusion

The study herein suggests that while changes in prostaglandins ($PGF_{2\alpha}$) and prostaglandin metabolites (PGFM, PGEM, PGIM) are associated with human labour, we did not identify any biomarkers that increased prior to labour onset and thus would have utility as predictors for term or preterm labour. However, we have identified additional eicosanoids present in at least half of our discovery cohort samples, that may play a role in pregnancy and labour. We have provided here the mass spectrometry lipidomics data and suggest that future studies should further investigate the role of these novel non-prostaglandin eicosanoids in pregnancy and labour.

## Supporting information

**S1 Appendix. ELISA and mass spectrometry data.**
(XLSX)

**S2 Appendix. Supplemental Fig 1. Urinary creatinine levels throughout pregnancy by gestational age at sample collection.**
(PDF)

**S3 Appendix. UCSD Lipidomics Core eicosanoid panel.**
(DOCX)

**S4 Appendix. Analyses excluding n = 4 participants with PPROM in the TPTL-PTD group.**
(DOCX)

**S5 Appendix. Supplemental Fig 2. Urinary prostaglandin metabolite levels not associated with cervical score in term or preterm pregnancy.**
(PDF)

## Acknowledgments

We thank our study participants and study support staff, in particular Jill Putnam. We would also like to thank Emma Walsh, whom we remember fondly.

## Author contributions

**Conceptualization:** Eilidh M. Wood, Stephen L. Wood, Donna M Slater.

**Data curation:** Eilidh M. Wood.

**Formal analysis:** Eilidh M. Wood.

**Funding acquisition:** Stephen L. Wood, Donna M Slater.

**Investigation:** Eilidh M. Wood, Kylie K. Hornaday, Matthew Newton, Melinda Wang.

**Methodology:** Stephen L. Wood, Donna M Slater.

**Project administration:** Stephen L. Wood, Donna M Slater.

**Resources:** Stephen L. Wood, Donna M Slater.

**Supervision:** Donna M Slater.

**Visualization:** Eilidh M. Wood.

**Writing – original draft:** Eilidh M. Wood.

**Writing – review & editing:** Kylie K. Hornaday, Matthew Newton, Melinda Wang, Stephen L. Wood, Donna M Slater.

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
