## [Decision Letter · Decision Letter 0]

Dear Dr. Slater,

Thank you for submitting your manuscript to PLOS ONE. After careful consideration, we feel that it has merit but does not fully meet PLOS ONE’s publication criteria as it currently stands. Therefore, we invite you to submit a revised version of the manuscript that addresses the points raised during the review process.

We look forward to receiving your revised manuscript.

Kind regards,

Godwin Upoki Anywar, BSc, Msc, PhD

Academic Editor

PLOS ONE

Journal Requirements:

Reviewers' comments:

Reviewer's Responses to Questions

**Comments to the Author**

1. Is the manuscript technically sound, and do the data support the conclusions?

Reviewer #1: Yes

Reviewer #2: Yes

2. Has the statistical analysis been performed appropriately and rigorously?

Reviewer #1: Yes

Reviewer #2: Yes

3. Have the authors made all data underlying the findings in their manuscript fully available?

Reviewer #1: No

Reviewer #2: Yes

4. Is the manuscript presented in an intelligible fashion and written in standard English?

Reviewer #1: Yes

Reviewer #2: Yes

Reviewer #1: Well-written article with logically conducted data analysis.

Unfortunately, the review of available current literature was not conducted correctly

1. the consent of the bioethics committee does not have a date of issue (day, month, year)

2. material and methods should provide the main metabolites that were analyzed (lipoxins, 5,8,12,15 HETE, leukotriene, thromboxane, protectins, maresins, resovins) or if only PG and their sunny please do not mislead in the description (also abstract)

3. the supplement may contain results indicating no differences between the remaining metabolites

4. the discussion should be rearranged and discussed to the already described fresh/new articles. Currently, out of 67 publications, only 10 items from the last 5 years. There are several hundred of them in the PubMed database alone

5. It is suggested to include new reports from the last 10 years

6. old literature should be removed and replaced 2-5, 7-9, 11-13, 21, 24, 27-35, 39-56, 58-59, 62-65

7. the summary right after the discussion without a description of the discussion line 281-282 is incomprehensible

8. the conclusion refers to 20 metabolites but it is not clear from the article which ones? it should be adjusted

best regards

Reviewer #2: This study investigates urinary prostaglandin metabolites as potential biomarkers for labor, with a focus on both term and preterm conditions. The research uses ELISA and mass-spectrometry to evaluate changes in prostaglandin levels across different gestational stages and labor conditions.

I would suggest some revisions as follows:

1)The threatened preterm labor (TPTL) group includes both preterm labor and preterm premature rupture of membranes (PPROM) cases. This might confound the results, as these conditions could involve distinct pathways. pls mention this in discussion limitations

2)Figures are informative but could benefit from improved clarity (e.g., larger font sizes for axes).

3)Some terms (e.g., "threatened preterm labor") should be clearly defined early in the manuscript for consistency and clarity.

4)If possible, reanalyze data by separating TPTL cases into labor-related and PPROM-related events to clarify molecular differences.

**Do you want your identity to be public for this peer review?** For information about this choice, including consent withdrawal, please see our Privacy Policy

Reviewer #1: No

Reviewer #2: No

---

## [Author Response · Author response to Decision Letter 1]

16 Jan 2025

Thank you for giving us the opportunity to submit a revised draft of the manuscript “Urinary prostaglandin metabolites as biomarkers for human labour: Insights into future predictors” for publication in PLOS ONE. We appreciate the time and effort that the reviewers have given to providing valuable feedback on our manuscript. Please see below for a point-by-point response to the reviewer’s comments and suggestions..

Reviewer 1:

Comment 1: the consent of the bioethics committee does not have a date of issue (day, month, year)

Response: We have included the date of issue of institutional ethics on line 99 (page 5).

Comment 2: material and methods should provide the main metabolites that were analyzed (lipoxins, 5,8,12,15 HETE, leukotriene, thromboxane, protectins, maresins, resovins) or if only PG and their sunny please do not mislead in the description (also abstract)

Response: We have included a description of the major classes of eicosanoids measured by mass spectrometry in the materials and methods section, lines 176-184, page 9. Details of all the eicosanoids measured are also included in S3 Appendix.

Comment 3: the supplement may contain results indicating no differences between the remaining metabolites

Response: All the raw data is present. We have not performed statistics on the mass spectrometry data in the supplemental file (Appendix 1) due to the small sample size for the discovery analysis combined with some eicosanoids that were below the limit of detection. We have included a statement of this in the statistical analysis section of the methods, lines 210-213, page 10.

Comments 4-6: the discussion should be rearranged and discussed to the already described fresh/new articles. Currently, out of 67 publications, only 10 items from the last 5 years. There are several hundred of them in the PubMed database alone. It is suggested to include new reports from the last 10 years. Old literature should be removed and replaced 2-5, 7-9, 11-13, 21, 24, 27-35, 39-56, 58-59, 62-65

Response: We have now included more up to date sources where possible, however in some cases the references cited are the most recent available primary articles on the subject, despite being published more than 20 years ago. New references are highlighted in yellow. Justifications for keeping select older references are detailed below.

- We have kept reference 9 (Karim et al., 1968) as it is the first report of the use of prostaglandins for induction of labour and provides important historical context surrounding the clinical use of prostaglandins in labour. However, we have also included two recent meta-analyses on the use of prostaglandins for labour induction.

- Reference 24 (Boeniger et al., 1993) is included in reference to a formula that was developed in said publication in 1993. We have also included a more recent citation that uses this formula for a similar purpose (MacPherson et al., 2018).

Comment 7: the summary right after the discussion without a description of the discussion line 281-282 is incomprehensible

Response: We have removed the “Summary” subheading from the discussion section (line 299 page 17).

Comment 8: the conclusion refers to 20 metabolites but it is not clear from the article which ones? it should be adjusted

Response: We have removed the reference to the 20 eicosanoids from the conclusion section to avoid confusion and have included a new table (Table 4 page 16) detailing the 20 eicosanoids that were detectable in at least half of the discovery cohort samples.

Reviewer 2:

Comment 1: The threatened preterm labor (TPTL) group includes both preterm labor and preterm premature rupture of membranes (PPROM) cases. This might confound the results, as these conditions could involve distinct pathways. pls mention this in discussion limitations

Response: Please see highlighted limitations section lines 387-393 (page 21).

Comment 2: Figures are informative but could benefit from improved clarity (e.g., larger font sizes for axes).

Response: Thank you for the suggestion. We have increased the font sizes of all the axes by at least 2 points.

Comment 3: Some terms (e.g., "threatened preterm labor") should be clearly defined early in the manuscript for consistency and clarity.

Response: We have expanded the description of threatened preterm labour in the introduction (lines 65-66, page 3) and have made the definition of threatened preterm labour more explicit in the methods section (lines 121-125, page 6).

Comment 4: If possible, reanalyze data by separating TPTL cases into labor-related and PPROM-related events to clarify molecular differences.

Response: We recognize this is a limitation of the present study, however it is not possible to analyze the TPTL cases separately by presence/absence of PPROM as there are only n=4 cases of PPROM within these groups. We have included an additional, separate analysis with the n=4 PPROM cases excluded in a new supplemental file (S4 Appendix) and have added a description of this analysis to the results (lines 234-238, page 12).

---

## [Decision Letter · Decision Letter 1]

Urinary prostaglandin metabolites as biomarkers for human labour: Insights into future predictors

PONE-D-24-52502R1

Dear Dr. Wood,

We’re pleased to inform you that your manuscript has been judged scientifically suitable for publication and will be formally accepted for publication once it meets all outstanding technical requirements.

Kind regards,

Godwin Upoki Anywar, BSc, Msc, PhD

Academic Editor

PLOS ONE

Additional Editor Comments (optional):

Reviewers' comments:

Reviewer's Responses to Questions

**Comments to the Author**

Reviewer #3: All comments have been addressed

Reviewer #4: All comments have been addressed

2. Is the manuscript technically sound, and do the data support the conclusions?

Reviewer #3: Yes

Reviewer #4: Yes

3. Has the statistical analysis been performed appropriately and rigorously?

Reviewer #3: Yes

Reviewer #4: Yes

4. Have the authors made all data underlying the findings in their manuscript fully available?

Reviewer #3: Yes

Reviewer #4: Yes

5. Is the manuscript presented in an intelligible fashion and written in standard English?

Reviewer #3: Yes

Reviewer #4: Yes

Reviewer #3: The revised version of the manuscript is improved and the authors made the appropriate changes suggested by the reviewers and replied properly to the comments and concerns.

There is a typo mistake: In Table 4 change Arachadonic acid to Arachidonic acid

Reviewer #4: Almost comments have been addressed.

There are some minor comments.

1) Line 511-513: Ref34 does not include the publication year or page number.

2) S3 Appendix: The statement "LOD = limit of detection" at the bottom of the table is unnecessary.

**Do you want your identity to be public for this peer review?** For information about this choice, including consent withdrawal, please see our Privacy Policy

Reviewer #3: **Yes: ** Paola Patrignani

Reviewer #4: No

---

## [Editor Report · Acceptance letter]

PONE-D-24-52502R1

PLOS ONE

Dear Dr. Slater,

I'm pleased to inform you that your manuscript has been deemed suitable for publication in PLOS ONE. Congratulations! Your manuscript is now being handed over to our production team.

Kind regards,

on behalf of

Dr. Godwin Upoki Anywar

Academic Editor

PLOS ONE